# Potential global impacts of alternative dosing regimen and rollout options for the ChAdOx1 nCoV-19 vaccine

Ricardo Aguas[1], Anouska Bharath[1], Lisa J. White [1], Bo Gao [1], Andrew J. Pollard [2], Merryn Voysey [2] & Rima Shretta [1✉]

The high efficacy, low cost, and long shelf-life of the ChAdOx1 nCoV-19 vaccine positions it well for use in in diverse socioeconomic settings. Using data from clinical trials, an individual-based model was constructed to predict its 6-month population-level impact. Probabilistic sensitivity analyses evaluated the importance of epidemiological, demographic and logistical factors on vaccine effectiveness. Rollout at various levels of availability and delivery speed, conditional on vaccine efficacy profiles (efficacy of each dose and interval between doses) were explored in representative countries. We highlight how expedient vaccine delivery to high-risk groups is critical in mitigating COVID-19 disease and mortality. In scenarios where the availability of vaccine is insufficient for high-risk groups to receive two doses, administration of a single dose of is optimal, even when vaccine efficacy after one dose is just 75% of the two doses. These findings can help inform allocation strategies particularly in areas constrained by availability.

[1] Nuffield Department of Medicine, University of Oxford, Oxford, UK. [2] Oxford Vaccine Group, Department of Paediatrics, University of Oxford, Oxford, UK. ✉email: rima.shretta@ndm.ox.ac.uk

As of August 5th 2021, over 200 million people have been diagnosed with COVID-19 worldwide, and in excess of 4.1 million confirmed deaths have been reported[1–3]. Vaccination is a critical strategy to control the spread of SARS-CoV-2, the virus that causes COVID-19, and to reduce the severity of symptomatic disease. At the time of the analysis, three vaccines had received emergency use authorization in the United Kingdom (UK). The developers of two of these vaccines have reported efficacies of 95% for their vaccines in their respective Phase 3 trials (Pfizer/BioNTech and Moderna)[4]. The third vaccine, ChAdOx1 nCoV-19, jointly developed by Oxford University and AstraZeneca, demonstrated an acceptable safety profile and efficacy against symptomatic COVID-19 (62.1% [41.0–75.7] vaccine efficacy against symptomatic infection)[5], with no hospital admissions or severe cases of disease reported in the intervention arm during Phase 3 trials conducted across three countries. This vaccine can be stored and distributed at 2–8 °C and is being made available at a lower cost than the other vaccines, making it suitable for global access, particularly in low-income and middle-income countries (LMICs)[5–7]. More recently, a single-dose vaccine developed by Johnson & Johnson has been shown to be 66% effective at preventing moderate to severe COVID-19, has been approved for use in the UK[8]; however, its role in the UK vaccination campaign is unclear given the proportion of people that have already received a first dose of a vaccine from another manufacturer. Countries where only a small proportion of the population have been vaccinated could consider this vaccine as an option to alleviate some logistical burdens associated with multiple dose vaccines.

While clinical trials have validated the efficacy of the ChAdOx1 nCoV-19 vaccine in reducing symptomatic infection, appropriate national vaccination strategies across the world must consider heterogeneity among populations as well as the diverse demographic and socioeconomic environments of affected countries. In particular, the younger population typically present in LMICs justifies the need to assess the effects of associated behaviors and health profiles on vaccine effectiveness. These countries exhibit competing health, social, and economic challenges owing to inadequate healthcare infrastructure and a high prevalence of immunocompromising and infectious diseases. In these settings, malnutrition and incidence of other infectious diseases might modulate vaccine immunogenicity in different ways when compared with individuals in more developed economies[9,10]. At the same time, many LMICs have been unable to secure enough vaccine doses from potential suppliers and thus are likely to have incomplete coverage of their populations, particularly in the short-term. The global COVID-19 vaccine alliance, COVAX, has pledged to procure and distribute vaccines equitably to LMICs; however, this will cover a maximum of 20% of the total population in each country[11]. Although the University of Oxford and AstraZeneca have made the largest supply commitment to LMICs at more affordable prices than other vaccine manufacturers, there is a need to evaluate the impact of a range of factors on vaccine effectiveness[12–14].

The UK government initially instituted a policy of administering the booster dose of the vaccine at up to 12 weeks following the initial dose, prompting a debate among scientists, manufacturers, and governments on the optimal dosing intervals for COVID-19 vaccines[15,16]. Given the global shortages in vaccine supply particularly in the short-term, a two-dose regime may not be feasible in settings with limited vaccine supply. This is particularly true in LMICs that have not been able to secure supplies of vaccines through advance market commitments (AMC) with manufacturers. The purpose of this analysis is to evaluate the efficacy of the ChAdOx1 nCoV-19 vaccine in countries with different demographic profiles, as a function of vaccine efficacy,

dosage regime (interval between initial and booster doses, or no booster at all), coverage, and immunity wane rate. Given the differences in healthcare infrastructure and vaccine access around the world, decision-makers should consider the effect of these factors on population-level impact to determine the most effective strategy for their context[14].

Where vaccination programs have begun, priority has initially been given to older age groups, individuals with co-morbidities, and frontline medical staff. The model developed therefore considers a simplified system, where the vaccine is delivered to age groups in descending order while supplies are available. Some studies have suggested that COVID-19 vaccines can limit the viral load in vaccinated individuals that become infected[17], which can translate into a transmission-blocking effect at the community level[18]. Unfortunately, this evidence has recently been confounded by the rise in incidence caused by the delta variant and suffers from a series of sampling and frame of reference issues, which result in the highest level of protection from the vaccine being found in the days immediately following vaccination—when antibody levels induced by vaccination are known to be negligible[6]. Thus, we start from the baseline assumption that indirect effects are negligible, and later present a sensitivity analysis exploring the impact of removing that assumption. Here we show that rapid delivery of the vaccine to the highest risk groups has the great impact on COVID-19 disease and mortality. In countries with insufficient availability for high-risk groups to receive both doses, administration of a single dose of vaccine is optimal. This effect is consistent even when vaccine efficacy after one dose is just 75% of the levels achieved after two doses. These findings offer a nuanced perspective of the critical drivers of COVID-19 vaccination effectiveness and can inform policies on allocation strategies, particularly in resource constrained environments.

## Results

**Epidemiological, logistical, and immunological factors**. We conducted an extensive initial sensitivity analysis to determine how the impact of rolling out a COVID-19 vaccination campaign in the UK depends on epidemiological, logistical, and immunological factors. The sensitivity of the modeled vaccine effectiveness to the variables explored is illustrated in Fig. 1. The figure illustrates that the prospects for vaccine impact are most sensitive to the number of vaccine doses available within 6 months, the speed of delivery within the same timeframe, and the vaccine efficacy (both the maximum efficacy post-dose two and the relative efficacy of dose one compared with dose two). Interestingly, for the same inputs, the median expected vaccine effectiveness is greater for deaths than it is for clinical cases.

From this first exploratory analysis one could immediately suggest that, for maximum effectiveness, a vaccine campaign should aim to vaccinate as many people as possible and governments/policymakers should therefore procure the maximum number of doses in the shortest time possible. These are by far, the two variables the model outputs are the most sensitive to, as illustrated in Supplementary Figs. 1 and 2. These figures also reveal an interesting interaction between the vaccine efficacy profile and an actionable decision of how the first available doses are to be distributed. In the frontloaded scenario, where the speed of vaccine delivery is maximal during the early stages of the 6-month vaccination campaign, a single-dose regimen is expected to perform significantly worse if vaccine efficacy post-dose one is 50% lower than vaccine efficacy post-dose two (top row). However, if the vaccine efficacy after both doses is the same (bottom row), a single-dose regimen can actually be preferable, especially if the number of vaccine doses available is small. In

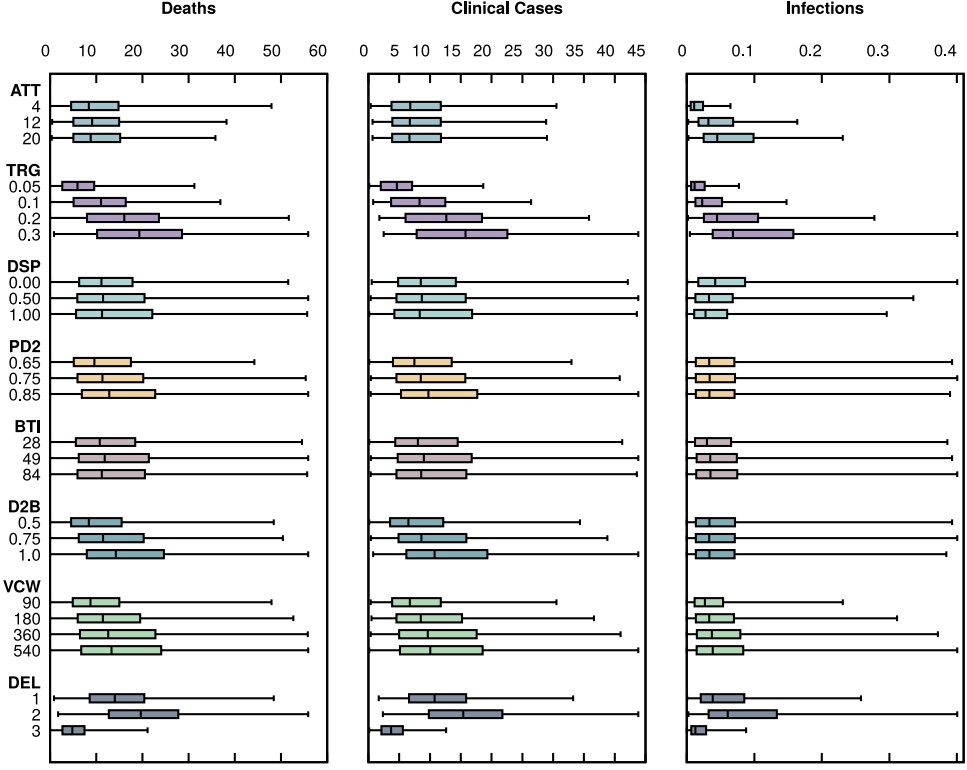

**Fig. 1 Overall sensitivity analysis of vaccine effectiveness (%), based on UK data.** The boxplots show the median and interquartile ranges of the predicted vaccine effectiveness (as a percentage) on each of the outcomes for specific parameters. They were generated by aggregating all model simulations for each of the parameters, with each boxplot summarizing the variance in predicted vaccine efficacy for all possible combinations of the other parameters. The middle line shows the median, the lower and upper hinges correspond to the first and third quartiles, and the whiskers extend to the 5th and 95th percentiles. The full list of explored parameters and their descriptions can be found in Table 1. ATT population attack size (as a % of the population), TRG vaccine allocation (% of the population during study period), DSP second dose administered (% of the vaccinated population administered a second dose), BTI interval between first dose and booster dose, PD2 vaccine efficacy after the second dose; D2B vaccine efficacy of the first dose compared with the second dose (%); VCW immunity wane rate (days following last dose), DEL vaccine delivery speed.

many settings, the cost, ease of implementation and the potential to cover a greater proportion of the population within a shorter time frame may take precedence over a small increase in efficacy.

**Interaction between vaccine efficacy and logistical variables.** The initial results prompted further investigation of the possible interactions and trade-offs between the vaccine efficacy profiles and logistical implementation variables. In this detailed analysis, the population attack size was fixed at 12%, delivery speed to frontloaded, and vaccine-induced protection to last 360 days. The results are summarized in Supplementary Fig. 3. As determined by the initial sensitivity analysis, vaccine effectiveness is quite sensitive to the number of available doses, the maximum post-dose two efficacy, and the efficacy of the first dose relative to the second.

Two interesting results pertain to the sensitivity of the model to changes in the dose number split and the interval in days between doses for the two-dose regimen recipients. While administering two doses to everyone, irrespective of all other variables, seems to be preferable to the single-dose option, the length of the whiskers suggests there might be a parameter space for which the single-dose option is optimal, as seen in Supplementary Figs. 1 and 2. Increasing the time interval between doses generally produces improved vaccine effectiveness, although a slight decrease in median effectiveness can be observed after an 8-week (56 day) interval. This is further investigated in Supplementary Figs. 4 and 5, revealing an interesting trade-off, with a large increase in predicted effectiveness after 7–8 weeks, followed by a small

decrease as the booster dose interval expands, but only if the efficacy of the first dose is low. If the efficacy of the first dose is similar to the efficacy of the second, increasing the interval between doses up to 12 weeks does not decrease vaccine effectiveness.

Interestingly, we find a non-linear increase in effectiveness for large values of dose availability, which can be explained by the markedly non-linear risk of severe disease and death with age. As the number of available doses increases, a larger proportion of the population will receive a vaccine dose. However, since vaccines are allocated in descending order of age in most countries, as a larger proportion of the population is reached, more and more low-risk individuals are vaccinated, for whom the vaccine accrued benefits are smaller and smaller. It is then advisable to investigate these relationships for different settings.

**Factors influencing decision-making.** We proceeded to investigate what factors could potentially influence the decision-making process regarding the distribution of doses during the first 6 months of vaccine program rollout. We thus evaluated the relative predicted effectiveness of the single-dose versus the double-dose regimen, for different countries with potentially different dose availability, and assuming different vaccine efficacy profiles (Fig. 2). For very high levels of dose allocation (high y-axis values), a two-dose regimen is clearly optimal. This starts to become less evident for scenarios, where the protection conferred by the first dose gets closer to the double-dose efficacy (moving right along the x-axis). Interestingly, when the number of

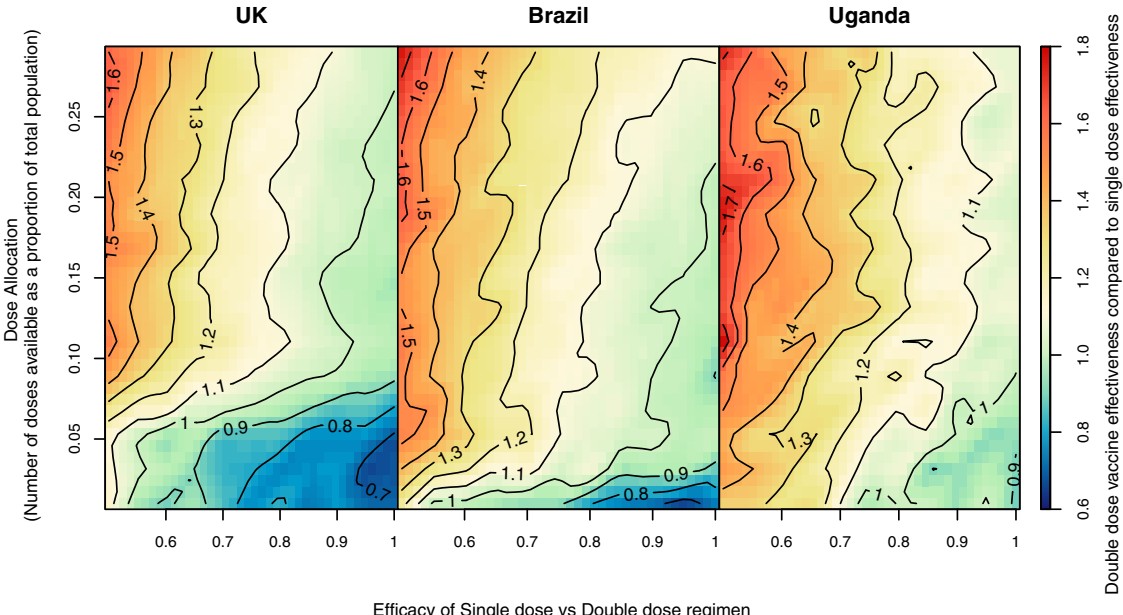

**Fig. 2 Optimal dose allocation.** The colored surfaces and respective contour lines indicate the ratio between the predicted vaccine effectiveness for a double-dose regimen vs a single-dose regimen, as indicated by the color-bar on the right. This ratio is a mean ratio, obtained by averaging out the ratios obtained in all model runs assuming the corresponding $x$ and $y$ parameter values and thus are not expected to be regular. Contour line 1 (thicker black line) indicates the parameter combinations for which there is no expected difference between giving everyone a single dose vs giving everyone two doses. For values greater than 1 (hot colors), a two-dose regimen is preferable, and for values less than 1 (cold colors), a single-dose regimen is preferable.

available doses is small, the single-dose regimen will become more effective than the double-dose regimen, as the thick black line is crossed. The parameter combinations defining the line where there is a shift in strategy positioning are country dependent, with countries with older populations having a larger parameter space in which a single-dose option is preferable, as shown in Fig. 3. These threshold allocation lines are generally robust to assumptions on vaccine indirect effects on transmission, especially for the most realistic range of a dose two efficacy boosting effect (0.7–0.8)[19] as shown in Supplementary Figs. 10 and 11.

## Discussion
The SARS-CoV-2/COVID-19 pandemic has created an unprecedented public health challenge, spurring a global race to develop and distribute viable vaccines. A vaccine that creates broad immunity against the SARS-CoV-2 virus could be the only effective means to control the pandemic and allow a return to "normalcy". To have a significant impact on the disease, a critical mass of the global population at risk will need to be vaccinated. However, many high-income countries have secured more than half of the available vaccine doses for themselves, leaving LMICs, which make up more than 85% of the global population, to find their own solutions[20]. To address the problem of equitable access, WHO, Gavi, and the Coalition for Epidemic Preparedness Innovations (CEPI) established COVAX, a global alliance that has pledged to pool investment and allocate and distribute COVID-19 vaccines equitably, particularly in LMICs[11]. However, COVAX is currently under-resourced and the doses secured are insufficient to achieve the coverage levels needed[21]. Supply constraints and new variants of SARS-CoV-2 are steering countries towards strategies that counter low access with dosing patterns or volumes to maximize the impact of the vaccines. Data from the ChAdOx1 nCoV-19 vaccine trials have allowed us to explore potential strategies to inform optimal allocation programs particularly in contexts, where the cost and logistics of implementing multiple doses within a short timeframe may be challenging.

Our findings indicate that vaccine effectiveness is dependent upon (i) the country context, which includes the demographic profile, the attack rate of the virus, and the amount of vaccine that is available (which influences the proportion of the population that is vaccinated); (ii) the characteristics of the vaccine, which include the efficacy of a single dose relative to a double dose and the waning of efficacy over time; and (iii) the proportion of the population receiving the second dose, the time interval between doses, and the delivery speed.

Our analysis demonstrates that in scenarios where the number of vaccine doses available is insufficient for the highest risk groups (aged > 65 years) to receive two doses, the allocation of a single vaccine dose is optimal. This effect is consistent even when the vaccine efficacy of a single dose is just 75% of the levels achieved after a double dose, until allocation drops to a population coverage of 10%, after which vaccinating only the high-risk individuals, with two doses, is more effective. In scenarios where the number of doses available to the country is sufficiently high, or if the relative single-dose efficacy is low (50% or less), providing a booster dose within 8 weeks would be preferable. Apart from these specific conditions, the results indicate that providing individuals with two doses of vaccine would have a similar effectiveness to the use of a single dose given to twice the number of individuals.

The speed at which the high-risk population is vaccinated greatly influences the expected vaccine effectiveness in preventing clinical cases and death. This is particularly true if the transmission rate is high, with faster vaccination reducing the number of infections in groups awaiting their first dose during the rollout. Distributing the vaccine very slowly provides an effectiveness of less than 10%, regardless of the number of doses and allocations. The impact of allocation on outcomes is also greater when the population is vaccinated rapidly over a 6-month period. In both of these scenarios, providing a single dose is preferable.

An interesting trade-off was found between the booster dose interval and the relative vaccine efficacy of a single dose. For vaccines with large differences between first and second dose

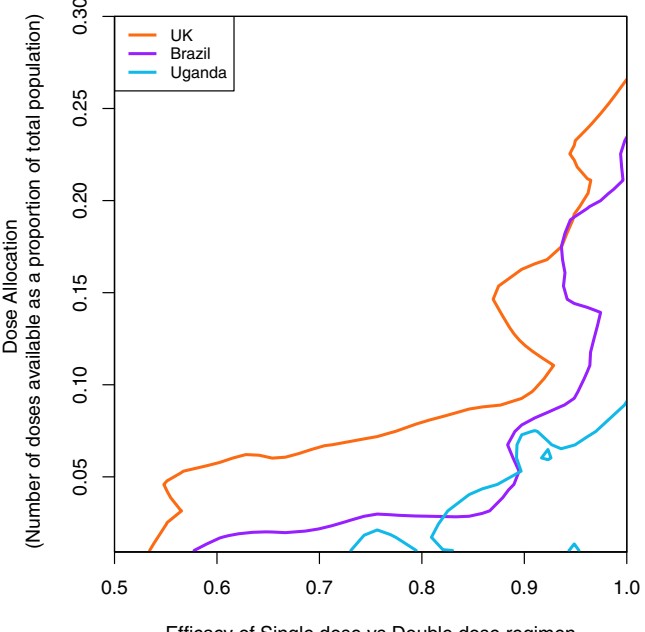

**Fig. 3 Dose allocation thresholds in different countries.** The figure illustrates the parameter combinations that define the allocation threshold above which a two-dose regimen would be preferred over a single-dose regimen—these are the thick black contour lines in Fig. 2. The areas enclosed by the curves are 16.5%, 8%, and 3.8% for the UK, Brazil, and Uganda, respectively, which correlates almost perfectly with the proportion of the population above the age of 65 years in those countries. To calculate these areas, we vectorize each contour line to obtain the y-values at which the contour line is crossed for evenly spaced x-values (taking the highest y-value for duplicates). The resulting vector is then run through a composite trapezoid rule algorithm that computes the approximate area under the curve (auc function in the MESS R package).

efficacy, delaying the booster dose interval past 8 weeks after the first dose was found to be detrimental. However, if a single dose provided at least 75% of the protection conferred by a double dose, delaying the booster dose interval to 12 weeks had a negligible impact on the number of cases and deaths. Given the similar reported efficacies of single and double doses of ChAdOx1 nCoV-19, a 12-week interval is the optimal scenario for this vaccine[19]. However, this finding may not be applicable to other COVID-19 vaccines.

These differences are more profound when considering the demographic characteristics of a population. In high-income countries, which have a larger older population (>65 years), a single-dose regimen will allow the vaccination of more individuals more quickly, with a correspondingly greater impact on cases and deaths. In the UK, the 6-month allocation threshold above which a two-dose regimen would be preferred was found to be about 16.5%.

The 6-month allocation threshold above which a two-dose regimen would be preferred is much lower in LMICs, mainly due to mortality in the younger population. In these contexts, decision-makers will need to consider the affordability, availability, and logistical constraints and feasibility of implementing a single or a double dose, the dosage intervals, and delivery speed. Most LMICs lack the digital databases necessary to manage patient data, reliably track vaccine inventories, keep track of who has received which vaccine, and inform people where and when they are due for a booster. Governments would also need to ensure that they reserve sufficient stocks to allow the

administration of booster doses. In these cases, a robust cost-benefit analysis of each option will need to be considered.

The dosing interval for COVID-19 vaccines has been a subject of debate among scientists, regulators, and governments around the world following the UK government's decision to prioritize administering the first dose of vaccine to as many at-risk people as possible and increasing the interval between the two doses to up to 12 weeks[15,16,22]. A one-dose vaccine regimen or a two-dose regimen with longer time intervals may be sufficient to reduce symptoms of COVID-19 in the most vulnerable individuals and ultimately slow the pandemic, given that the time difference between first and second doses was shown to have a negligible effect on overall vaccine effectiveness (clinical cases, infections, and deaths). Indeed, a recent WHO notification stated that some countries are facing "exceptional circumstances" and may want to delay second doses to "allow for a higher initial coverage". Other exceptional circumstances may involve trade-offs around the relative size of the highest risk population in a country and the currently unknown potential for a vaccine to reduce transmission, which may lead to some countries targeting high-contact groups to benefit from any potential indirect effects.

The thresholds considered in this work will differ depending on country contexts and vaccine efficacy related parameters. We have included the potential impact of age structures and dose availability on the policy implications of different vaccine schedules but should emphasize that further research on specific epidemiological and health system contexts, especially in LMICs, is needed. In particular, the relative proportion of high-risk groups versus high-incidence groups, familial structures, mixing patterns between high-risk and high-incidence groups, vaccine access equity across risk groups, and vaccine regulatory processes leading to approval and recommendation, stand out as critical elements. A study in Thailand inferred that vaccinating the high-incidence group with an infection-blocking vaccine (>50% protection against infection)[23] could provide enough indirect effects for that strategy to be preferred over vaccinating the same number of high-risk group individuals. In our sensitivity analysis (Supplementary Fig. 11), we find that even in "older" countries like the UK, an infection blocking vaccine could reshape the parameter space where a double dose strategy is optimal. If individuals with a second dose of vaccine have up to a 50% chance of transmitting the virus, for each 10% increase in vaccination coverage there is a 5% decrease in overall transmission; disproportionately increasing the value of fully vaccinated persons in the population immunity landscape. Thus, as the vaccine effect on transmission increases, fully vaccinated individuals acquire more value, and thus the two-dose strategy is preferred over the single-dose strategy for the majority of the parameter space.

In countries with a higher vaccine availability than that considered in this analysis, a vaccine providing a large impact on onwards transmission could be extremely impactful given very rapidly to the subset of individuals driving the transmission. Similarly, smaller countries with robust Expanded Program on Immunization (EPI) programs may be able to rapidly roll out the vaccine to a higher percentage of their population compared with some larger countries. Thus, country-level idiosyncrasies and contexts should determine the optimal implementation strategy.

Recent discussions have focussed on the potential population level impact of an additional third booster dose of the vaccine. The lack of clinical data on the potential additional benefits of a third dose warrants further research, especially given the uncertainty around the duration of the vaccine's protective efficacy following the second dose. High-income countries that started vaccination campaigns over 6 months ago are beginning to

distribute third doses, perpetuating the inequity in global vaccine distribution. This has prompted the WHO to issue a call to prioritize the maximization of vaccine coverage globally in detriment of additional booster doses[24]. This could be achieved by following the allocation metrics proposed here, ensuring that LMICs receive the threshold vaccination doses that maximize impact on clinical disease as indicated by Figs. 2 and 3. More critically, new evidence indicating that single dose vaccine efficacy is particularly sensitive to the presence of the delta strain[25], unlike post-dose two efficacy, suggests that in areas where the delta variant is highly prevalent, a single dose might confer poor protection against clinical disease (left hand side of Figs. 2 and 3). Single dose vaccines would therefore better serve those countries with the lowest current vaccine coverage thus enabling a faster protection of the high-risk populations across the globe, providing they are efficacious against the delta variant.

Published clinical data were used to inform the parameters used in the model described in this paper. These data provide an aggregate efficacy of the ChAdOx1 nCoV-19 vaccine among people of a wide range of ages living in different countries. However, there were limited data available for assessing the effects of certain parameters (such as the effect of the dosing interval on post-dose two or dose three efficacy) on vaccine efficacy, which reinforces the need to conduct the post hoc exploratory analysis presented here. New data is continuously being generated from long-rolling ChAdOx1 nCoV-19 vaccination programs with long delays between doses (12 weeks), now showing a rapid drop in mortality even after the first vaccine dose[26], supporting results presented here, and decisively aiding vaccine allocation decisions in other countries.

This analysis demonstrates that in scenarios where the number of vaccine doses available is insufficient for the highest risk groups (>65 years of age) to receive two vaccine doses, allocation of a single vaccine dose to twice the number of individuals or extending the time interval between doses may be more optimal strategies. In contexts without supply constraints, or if the single-dose efficacy is low, providing a booster dose would be preferable. Apart from these specific conditions, the results indicate that providing individuals with two doses of vaccine would have a similar effectiveness to the use of a single dose in twice the number of individuals. In an ideal world, decisions about vaccination strategies would be made within the exact parameters of the trials that have been conducted. However, the limited availability of resources, and specific country contexts, may require decision-makers to consider alternative strategies. Given the recent discussions and approval of a third booster dose in some countries[27], there is an urgent need to conduct further analyses on optimal dosage strategies incorporating immunity waning (time and shape distribution) and evaluate how these differ with one-dose, two-dose and three-dose regimens in a variety of demographic settings.

## Methods

The methodology employed was very specifically tailored to the research question and its context. Vaccine production rates are always going to be insufficient to meet the demand generated by a global pandemic. In a context of limited vaccine dose availability, it is imperative to prioritize those individuals who would yield the greatest epidemiological benefit. Assuming the most pressing need is to reduce hospitalization rates and deaths, the initial targeting of those at higher risk for these outcomes seems logical, given that the alternative of immunizing sufficient people at lower risk for the indirect benefits to outweigh the direct benefits of a vaccine targeted at those at higher risk is not feasible with the number of vaccines available in the short-term. Even the UK, where mass production of the AstraZeneca vaccine has enabled 20% of the population to be vaccinated within 3 months, opted to prioritize the high-risk groups (those aged more than 65 years), partially because of the uncertainty around vaccine efficacy against infection. The ChAdOx1 nCoV-19 vaccine clinical trials were the only Phase 3 trials in which infection was evaluated as an outcome. No evidence was found for a transmission reduction effect (VE = 3.8% [−72.4 to 46.3])[5], but important questions were raised about how to allocate a limited number of doses to optimize the impact on symptomatic disease, given that a single-dose regimen could offer prolonged protection and thus a delay of the second dose could be warranted.

We began from the premise laid out above and implemented an individual-based, age-dependent, static transmission model to predict the number of infections, clinical cases, and deaths expected to occur within 6 months of vaccination program rollout. Individuals are simulated as autonomous systems, each with a set

---

**Box 1. | Consider a hypothetical unvaccinated population that we suppose would experience 100,000 SARS-CoV-2 infections over the following 6 months. Policymakers faced with such a prospect could opt for one of two alternative options (with very different direct benefit outlooks) for allocating the limited number of vaccine doses available to them**

Option 1: Vaccinate high contact groups (aged 25–40 years) (20.3% of the population, 36% of all infections, 10% of all hospitalizations, 1% of all deaths)[1]
Predicted infections:
100,000*36%*[1-vaccine direct effect on infection] + 100,000*64%*[1-vaccine indirect effect on infection]
Predicted hospitalizations:
Predicted infections*IHR*10%*[1-vaccine direct effect on hospitalization] + Predicted infections*IHR*90%*[1-vaccine Indirect effect on hospitalizations]
Predicted deaths:
Predicted infections*IFR*1%*[1-vaccine direct effect on deaths] + Predicted infections*IFR*99%*[1-vaccine indirect effect on deaths]
Option 2: Vaccinate high risk groups (aged > 65 years) (18.77% of the population, 10% of all infections, 66% of all hospitalizations, 90% of all deaths)[1]
Predicted infections:
100,000*10%*[1-vaccine direct effect on infection] + 100,000*90%*[1-vaccine indirect effect on infection]
Predicted hospitalizations:
Predicted infections*IHR*66%*[1-vaccine direct effect on hospitalization] + Predicted infections*IHR*34%*[1-vaccine indirect effect on hospitalizations]
Predicted deaths:
Predicted infections*IFR*90%*[1-vaccine direct effect on deaths] + Predicted infections*IFR*10%*[1-vaccine indirect effect on deaths]
Taking the direct effects from the ChAdOx1 nCoV-19 vaccine trial 5, we can easily extrapolate what the vaccine indirect effects would have to be for Option 1 to prevent more deaths than to Option 2 by comparing the predicted deaths for the 2 options above and solving that inequation for vaccine indirect effect on deaths, $X$:
$0.01*(1 − 0.85) + 0.99*[1 − X] < 0.9*(1 − 0.85) + 0.1*(1 − X)$
$X > 0.85$
Thus, vaccines targeting high-contact groups would have to provide indirect effects of 85% to prevent of the same deaths predicted to occur when targeting the highest-risk group with a direct vaccine effect against death of 85%.

---

of attributes, informing their serostatus, vaccination uptake history (number of doses and dosing interval), and age. Box 1 details how the dynamical processes inherent to disease transmission and vaccination outcomes are considered in the model. It clearly shows that a vaccine with an immunological profile like the ChAdOx1 nCoV-19 vaccine should prioritize the elderly population in order for deaths to be minimized with the least number of doses and requiring limited indirect benefits. Supplementary Fig. 7 demonstrates clearly how the predicted reduction in deaths resulting from targeting high risk groups is critically dependent on direct effects (outcome changes along the $x$-axis), whereas vaccinating the same amount of 25–40 year-olds would yield a lower impact on deaths that instead is more reliant on indirect effects (changes along the $y$-axis). This much was clear to policy makers wishing to decrease COVID-19 mortality and morbidity as quickly as possible, as reflected by vaccine rollouts all developed countries. The lingering question of how much transmission reduction we could expect to achieve at the population level when vaccinating the highest-risk group is addressed in Fig. S8. The reach of vaccination impact on viral transmission can only be thought of as proportional to the contribution of the targeted group to overall transmission. Given that in the high-risk group we only find 10% of all infections, even though it constitutes ~19% of the population, we can guess there would be a suboptimal reduction in community transmission when vaccinating this group only. Assuming that indirect effects are proportional to the distribution of infections on the different target groups, we obtain an expected decrease in total population infections lower than 1% for all combinations of direct and indirect vaccine protection explored here when vaccinating the elderly population (Fig. S8).

Thus, we can confidently ignore indirect effects for the vaccination strategies explored in this paper and adopt a static framework to compare vaccine delivery schedules.

**Transmission and clinical cascade.** The spread of COVID-19 is sensitive to the underlying network of contacts between infectious and susceptible individuals in their various societal spheres (home, work, public transport, etc). For a given population, we can summarize the number of contacts per day as an age-dependent force of infection $\lambda(a)$, i.e., a daily risk of acquiring an infection given age $a$. The age-dependent risk of infection can then be defined as:

$$\lambda(a) = k_\lambda \sum_{w=1}^{N} c_{aw} \frac{\sum_{v=1}^{N} c_{vw}}{\sum_{v=1}^{N} \sum_{w=1}^{N} c_{vw}}, \tag{1}$$

where $c_{aw}$ is the daily number of contacts between people of age groups $a$ and $w$, $N$ is the total number of age categories, and $k_\lambda$ is the overall daily risk of infection (which is informed by the number of infectious people in the population). Here, we decided to simplify the transmission process by making the daily risk of infection constant over time, thus having a static transmission model. Initially, we performed a calibration process whereby we ran the model without vaccination thousands of times, each time assuming a different daily risk of infection parameter. We then

selected those parameter values that gave us the desired attack rates over a 6-month period as specified in Table 1. By attack rate we mean the proportion of the population that is newly infected during those 6 months.

The risk of developing severe disease and possibly dying as a consequence of infection was informed by age-dependent infection hospitalization (IHR) and hospitalization fatality (HFR) ratios, published in ref. [28]. Thus, the modeled daily risk that an individual will develop severe disease is given by $\lambda(a)$IHR$(a)$, whereas the risk of dying is approximately $\lambda(a)$IHR$(a)$HFR$(a)$. The timing of these events and the lag between infection and clinical outcome are not relevant, as we are only making comparisons between synthetic populations, as detailed below.

**Vaccination delivery and vaccine efficacy.** Different vaccine dose allocation schemes were simulated, by limiting the number of doses distributed in 6 months, as well as allowing for different dosing intervals (delaying the second dose) and dosing splits (giving one dose vs. two doses). The allocation of doses was always prioritized to the oldest age groups. Individuals were assigned vaccine doses in descending order of age until the maximum number of doses had been allocated.

Given a fixed number of available doses, one can calculate the target recipient population by looking at the dose-split proposal. If all doses are given as single doses in 6 months, then the target population for vaccination is equal to the number of available doses. At the other extreme, where all vaccinees receive two doses, the number of recipients would be half the number of available doses. Within the group that is meant to receive two doses of the vaccine, a 5% dropout rate (vaccine refusal) was imposed, and a range of booster dose intervals was explored—Supplementary Fig. 9.

We implemented three different logistical implementations of a vaccine campaign rollout: constant effort, frontloaded, and backloaded. The distinction was in the speed at which the target population received vaccine doses during the initial 2 months—Supplementary Fig. 6. As individuals were assigned a vaccine, the number of doses received would be determined by a draw from a uniform distribution according to the desired dose split. Individuals given two doses would be assigned a booster dose interval following a beta distribution with $\alpha = 0.15$ and $\beta = 0.95$.

Although vaccine efficacy was explored in the sensitivity analyses presented here, we centered the explored ranges around the point estimates presented in ref. [19]. Vaccine efficacy, $V_e(t)$, was treated as a direct modulator of the risk of infection, clinical disease, and death; it was then defined for each individual, at each timestep of the simulation, as:

$$V_e(t) = V_i^j e^{-\delta t}, \tag{2}$$

where V is the vaccine efficacy in an individual with baseline status $i$ that received dose number $j$ a $t$ number of days ago, while $\delta$ is the rate of loss of vaccine-induced immunity. Baseline status is a binary variable reflecting the susceptibility status (immune or non-immune) of each person before the vaccination campaign starts.

Throughout this paper, we present a sensitivity analysis of the post-dose two maximum efficacy, the relative efficacy of dose one vs dose two, and the booster

---

## Table 1 Model parameters.

| Parameter | Model term | Range | Description |
|---|---|---|---|
| Population attack size (% of the population) | ATT | (4, 12, 20) | This is the percentage of the population infected within the 6-month study period |
| Vaccine allocation (% of the population during study period) | TRG | (5, 10, 20, 30) | Allocation range was based on the assumed administration speed. Using current data, we assumed that higher-income countries could reach a maximum speed, allowing 30% coverage of the population within 6 months |
| Second dose administered (% of the vaccinated population administered a second (booster) dose) | DSP | (0, 25, 50, 75, 100) | This is the percentage of the vaccinated population that are administered a second (booster) dose |
| Interval between first dose and booster dose | BTI | (4, 7, 12 weeks) | The interval between doses can affect vaccine efficacy; the range chosen was based on available clinical trial data |
| Vaccine delivery speed | DEL | (fixed, frontloaded, backloaded) | The speed of vaccine delivery to the population—see Supplementary Fig. 6 |
| Vaccine efficacy after the second dose | PD2 | (65, 75, 85) | Maximum efficacy following the second dose |
| Vaccine efficacy of the first dose compared with the second dose (%) | D2B | (50, 75, 100) | Effect of the first dose compared with the second dose |
| Immunity wane rate (days following last dose) | VCW | (90, 180, 360, 540) | Vaccine protection decay post-last dose |
| Overdispersion of vaccine impact on transmission | $\sigma$ | 4 | Sets the magnitude of the daily fluctuations in vaccine impact on transmission if RT is larger than 0 |
| Vaccination impact on the effective reproduction number (%) | RT | (0, 25, 50) | Maximal transmission potential reduction following a vaccine second dose. In the main analysis it is assumed to be 0% and is varied in a supplementary sensitivity analysis |

dose interval. We also imposed a stepwise increase in post-dose two vaccine efficacy across an 8-week booster dose interval, as observed in the clinical trial[19]. This means that giving the second vaccine dose less than 8 weeks after the first dose will result in a 25% lower post-dose two efficacy relative to the maximum assumed vaccine efficacy. While doing so, we constrain vaccine efficacy against clinical disease to be the same as that against death, while vaccine efficacy against infection is fixed at 5%[5] in a first instance. In a separate sensitivity analysis, we explore how the results presented in the main text are sensitive to vaccine indirect effects on transmission. To do so, we calculate a daily modulator $\left(\hat{vb}\right)$ of the risk of infection which accounts for vaccine impact on onward transmission that depends on the proportion of people in the population with $j$ doses of the vaccine and the relative impact conferred by each dose. In plain terms, the overall impact on transmission of vaccinating a proportion of the population should be equal to the mean decrease in transmission across all individuals, resulting in:

$$\hat{vb}(t) = 1 - \frac{\sum_{i=0}^{i=N} v_{\text{impact}}^{j(i)}}{N}.$$

Consistent with the remaining vaccine efficacy parameters, we assumed there is a boost in vaccine impact on transmission with an increasing number of doses (given by the parameter D2B in Table 1). To reflect how the network of effective infectious contacts includes different proportions of vaccinated people each day, we assume that the mean impact on transmission changes daily. This is done by sampling a population level impact on transmission $vb$ assuming a Beta distribution with overdispersion $\sigma$:

$$vb(t) \sim \beta\left(\hat{vb}(t), \sigma\right)$$

This parameter $vb$ changes daily as more people get vaccinated and modulates the population force of infection laid out above ($\lambda(a)$) directly, where:

$$\lambda(a, t) = \lambda(a) \cdot vb(t)$$

**Vaccine effectiveness**. The vaccination campaign population impact is referred to throughout as vaccine effectiveness and was defined as:

$$V_{\text{eff}} = 100 \frac{AR_v - AR_u}{AR_u}, \tag{3}$$

where $AR_v$ is the attack rate (over 6 months) in the vaccinated population, and $AR_u$ is the attack rate in a population that mirrors the vaccinated population in all aspects except vaccination. We thus have a pair of populations for each parameter set in our analyses and calculate the expected vaccine effectiveness for each parameter set as the relative difference in occurrence of each of the disease end-points (infection, clinical symptoms, and death).

**Reporting summary**. Further information on research design is available in the Nature Research Reporting Summary linked to this article.

## Data availability

The data used to inform this analysis have been published and are available in the public domain (https://doi.org/10.1016/S0140-6736(21)00432-3)[19]. Data on participants from the clinical trials can be obtained upon request from the Oxford Vaccine Group (merryn.voysey@paediatrics.ox.ac.uk) when the trials are complete.

## Code availability

The code data generated in this study have been deposited in the github database https://github.com/ricardoaguas/como-ChAdOx1-vaccine- available at https://doi.org/10.5281/zenodo.5522794.

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

## Acknowledgements

L.J.W. is funded by the Li Ka Shing Foundation. R.A. is funded by the Bill and Melinda Gates Foundation (OPP1193472). The Covid-19 Modelling Consortium has support from the Oxford University COVID-19 Research Response Fund (ref: 0009280). The authors have not been paid to write this article by a pharmaceutical company or other agency. The funders played no role in the design or outcomes of this work. The authors are grateful to Adam Bodley for proofreading the final draft.

## Author contributions

R.A., A.B., L.J.W. and R.S. conceived the paper and contributed to the analysis. R.A. developed the model and wrote the code. B.G. ran the PSA and contributed to data processing. M.V. and A.P. provided the data, discussed the analysis, and commented on the draft. All authors have read, contributed to, and approved the final version of the manuscript.

## Competing interests

The authors declare no competing interests.
