## [Peer Review File · Nature Communications]

REVIEWER COMMENTS

Reviewer #1 (Remarks to the Author):

Comments for Author

This paper uses static agent-based transmission modelling to explore circumstances under which allocating of a single or a double dose of the ChAdOx1 nCoV-19 vaccine will have higher vaccine effectiveness across different settings. The authors use data from the clinical trials of this vaccine, and combine it with a static transmission model to explore how vaccine effectiveness is affected by a) the characteristics of the vaccine quantified by the ratio of the efficacy of single dose vs double dose of the vaccine and the efficacy waning over time and b) the administration of the vaccine including the proportion of people receiving the second dose, the delivery speed and the time interval between the two doses and c) the settings profile defined by the demographic profile – specifically the proportion of the population that is 65+-, the vaccine availability and the attack rate of the virus in the setting. The results showcase the different trade-offs between a)-c) need to be considered for decision making around how this vaccine roll-out is delivered across different countries. For example, one of the main analysis scenarios suggests that if the number of doses available is not sufficient to cover all 65+ (considered people at highest risk) with two doses, then allocation of a single vaccine to twice as many individuals or extending the time interval between doses may be necessary. This is dependent how effective I the first dose vs the second dose and the results are not surprising. While, this is a very valuable study, I have some reservations in accepting this manuscript as it is and would suggest some changes are necessary before it is accepted.

Major comments:

- 1) The simulations at the core of this work is a static (time-invariant) transmission model. I feel that having a static model is not sufficiently robust for this analysis and suggest using a dynamic agent-based model that simulates the individuals with infection attributes (susceptible, infectious, hospitalised etc) that change over time, in addition to the belonging in different society layers (home, work, etc). The force of infection λ will then not only depend on age but also on time and also possibly on the viral profile- as for example captured within the dynamic agent-based model, for example. <https://www.medrxiv.org/content/10.1101/2020.05.10.20097469v3>
- 2) The vaccine drop out rate modelled is 5%. I would like to see a sensitivity analysis on this value and especially in light of possible vaccine hesitancy towards ChAdOx1 nCoV-19 vaccine in different settings.
- 3) Important characteristics of a vaccine against COVID-19 is the reduction in severity from vaccination – quantified by reduced hospitalisations and deaths- but also the impact of onwards transmission. The authors here assume that the vaccine efficacy against infection is 5% and they do not untangle the relationship between vaccine allocation and effectiveness of the vaccine on transmission. Decisions of rolling out vaccination in younger age groups, or in settings with younger population, is crucially depended on this and any informed evidence on vaccine allocation has to address this issue. My suggestion would be to include sensitivity analysis on this parameter and reproduce Figures 1-3 for different values of this parameter.

Minor comments

- 1) Please give more detail on why the booster dose interval is following a specific beta distribution – and explain why variation of this is not considered in additional sensitivity analysis.

Reviewer #2 (Remarks to the Author):

This is an interesting paper that explores some trade-offs with dosing strategies for the ChAdOx1 vaccine. The authors find that, under conditions of constrained supply, in some situations it is optimal to give a single dose to as large a section of the population as possible (starting from elderly and working down) and in other situations prioritising two doses works better. This trade-off depends on the size of the supply population age structure in a relatively simple way -- clearly explained in the discussion.

While I suspect that the trade-off is described in a way that is qualitatively accurate, I would take the specific numbers obtained with a grain of salt for methodological reasons as I will explain. The identification of this tradeoff and its qualitative nature is a valuable contribution and I therefore recommend acceptance subject to revisions.

The authors explain, "the methodology employed was very specifically tailored to the research question and its context." However no source code is available for inspection from what I can tell. This is a serious issue for evaluating the model, for transparency and reproducibility. I always recommend against publication of papers based on models where the source code is not publicly available.

The authors use an individual based model which in principle could be an unconstrained computer program that does anything, however there is a hint that it is much simpler than that: individuals are described by a tuple of attributes which are modified in a uniform way. This is good because it suggests that the model is just a stochastic rewriting system perhaps equivalent to a rule-based model or a coloured Petri net. I would find this more convincing than a tailored, bespoke, ad-hoc model because it means that we know how to understand the conceptual framework where it sits.

The equation in "transmission and clinical cascade". Please number your equations. λ is a function $\text{age} \rightarrow \mathbb{R}$. However age (a) does not appear on the right hand side. I think it perhaps corresponds to the "I" subscript in the numerator but as written this appears to be a constant vector-valued function of age which is clearly not what is intended.

However, this definition of λ is motivated by computational simplification because the simulation is expensive. It is unclear why the simulation should be so expensive. In particular, the authors say that the daily risk of infection is a function of time and is adjusted (how?) such that the correct attack rate is obtained. In the absence of a transmission model, what is the correct attack rate? How is this obtained? Surely it is itself influenced by vaccination which reduces the chance of infection. The mechanism here is unclear and is the main reason why I say that quantitative results obtained in this way should be taken with a grain of salt.

Is it perhaps feasible, given that parameter regimes of interest have been identified using the relatively inexpensive model, to then explore those interesting regimes with a more fully-fledged model containing transmission? That would be my preferred approach over a static model.

Vaccine efficacy decays exponentially (next equation). This doesn't seem biologically realistic. At $t = 0$ I would expect efficacy to be zero and for it to increase to some maximum over a period of time and then to reach a plateau (probably not a sharp peak) and thereafter to slowly decay. This could matter because the time to reach the plateau is on a similar scale to the serial interval.

Ideally I would like to see the effect of the vaccine on transmission incorporated here though I

understand if the authors feel that it is out of scope. For countries with high rates of vaccination this is clearly important as we are already seeing in the USA and the UK. For countries that are severely supply constrained such as the LMICs that are really the main focus of this paper, this effect may not be so strong. At a minimum this should be discussed.

The observation of the threshold value for choosing between strategies corresponding to the curve whose integral is the proportion of the population over 65 mentioned in the caption of Fig 3 could usefully be emphasised. This seems to be the main result, actually, and it is mentioned almost in passing at the end of the Results section. However, it is not clear to me how the integral is defined because the curves are not functions (there are multiple "y" values for some "x"). Please clarify this and explain.

Reviewer #3 (Remarks to the Author):

This article uses static transmission modelling to predict the impact of ChAdOx1 vaccine rollout in limited-resource scenarios, altering parameters such as the timing of vaccine rollout and administering one vs. two doses of vaccine. This work is timely, important, and makes an important acknowledgment of the hard decisions surrounding vaccine allocation that will continue to be made in the coming months.

The authors have taken the trouble to quantify an answer for vaccine allocation that may largely be intuitive: that the greatest number of COVID-19 cases are averted with high vaccine coverage that is rolled out as quickly as possible. Additionally intuitive may be the finding that, as long as VE for one dose is at least 51% of the VE for two doses, a greater number of illnesses may be averted if twice the people receive half the doses. It is reassuring to see that these back-of-napkin, informal calculations hold up when additional, formal model considerations are added. I believe it will be helpful for the authors to more clearly highlight circumstances under which these "intuitive" findings could be violated. If there is time and space, it would be extraordinarily helpful to identify how these assumptions could change under additional scenarios, potentially including:

1. ChAdOx1 VE against circulating variants of concern, especially in LMIC settings where VOCs may cause the majority of cases
2. National regulatory agencies' decision-making process regarding ChAdOx1 vaccine safety in younger populations, and how this could influence age-based prioritization and/or speed of rollout
3. Alternate scenarios in which (usually younger) healthcare workers are prioritized for vaccination alongside older adults; this may be of special interest given SAGE recommendations for vaccine prioritization and given new safety outcomes of interest in Europe

Additional minor comments are listed below.

- Page 4, paragraph 1: It would be helpful for the purposes of contrast to also list the estimated efficacy for ChAdOx1, since it is listed for the Pfizer and Moderna vaccines, and to specify the outcomes for which efficacy was measured (for all vaccines)
- Page 4, paragraph 2: As a point of clarification, when the authors say "suffer complex vaccine responses", are they referring to vaccine safety/adverse event outcomes?
- The authors note a hospitalization fatality rate; perhaps they could note which proportion of COVID-19 deaths in low-resource settings they assume to occur outside the hospital.
- Page 9, paragraph 5: By "an individual with baseline status i ", do the authors mean "baseline vaccination status"?
- At the end of the discussion section, it could be helpful to briefly mention which additional observational data (e.g., VE estimations from Canada which will now include a 4-month delay between dose 1 and dose 2) will be helpful to update these calculations

- In Figure 1, is the top bar x-axis representing VE? If so, it would be helpful to provide that information in the axis title or labels. Similarly, it can be challenging to flip back and forth between Table 1 and Figure 1; it would be helpful to include the abbreviations in row titles in Figure 1 as a footnote as well.
- In Figure 2, if I understand the figure correctly, I recommend the vertical bar on the right side be removed and rotated 90 degrees so that it reads more as a “legend” than as a secondary y-axis. (Again, if I do not understand correctly, my apologies.)
- In Figure 2, it is not clear to me what the decimal represents for “dose allocation”—does this represent the percentage of the population vaccinated, the percentage of total doses that are given in a 6 month time frame, or other?

REVIEWER 1 COMMENTS:

Major comments:

- 1. The simulations at the core of this work is a static (time-invariant) transmission model. I feel that having a static model is not sufficiently robust for this analysis and suggest using a dynamic agent-based model that simulates the individuals with infection attributes (susceptible, infectious, hospitalised etc) that change over time, in addition to the belonging in different society layers (home, work, etc). The force of infection λ will then not only depend on age but also on time and also possibly on the viral profile- as for example captured within the dynamic agent-based model, for example. <https://www.medrxiv.org/content/10.1101/2020.05.10.20097469v3>***

Author response: We have explained in the paper that the methodology employed was specifically tailored to the research question (prioritising single-dose vs double dose vaccine coverage) and its context (using the ChAdOx1 nCoV-19 vaccine in a top-down age priority schedule). The ChAdOx1 nCoV-19 Phase 3 vaccine clinical trials were the only Phase 3 trials to present infection as one of the tracked outcomes. No evidence was found for a transmission-blocking effect (VE = 3.8% [-72.4 to 46.3]) both overall and at any stage of follow-up. Given the broad confidence intervals, there could be an argument that some indirect effects might be accrued. We demonstrate in the methods section (Box 1), that given the low values of coverage explored here, the vaccine would need to have a very high infection blocking profile for indirect effects to be significant in our simulations, which would be against the evidence produced by the clinical trials. Also, there is no scope to vaccinate younger individuals (where indirect effects could play a larger role) since the vaccine prioritisation schedule was fixed to mimic the UK vaccine rollout (which is similar to most HICs), and thus, the potential for reductions in transmission were assumed negligible throughout our simulations. This is further elaborated at the beginning of the methods section and illustrated with two supplementary figures. In addition, given that vaccine production rates are likely to be insufficient to meet the demand generated by a global pandemic in the short term, even if the vaccines did have some infection prevention properties, vaccine coverage would not be able to reach the high rates needed to achieve significant herd effects. We therefore purposefully implemented an individual-based, age-dependent, static transmission model to predict the number of infections, clinical cases, and deaths expected to occur within the first 6 months of the vaccination programme rollout.

A recent paper¹ employed a dynamic model to explore optimal vaccination schedules in the UK specifically and concluded that even under the most optimistic scenario for protection against new infections (85%), vaccinations would

not reduce the reproduction number R below 1, and that the most beneficial strategy would always be to vaccinate the elderly first, regardless of indirect effects. We have now provided figures that highlight the analytical reasoning behind this and justify the methodology employed and choice of model given the context and vaccination characteristics.

¹ Moore S, Hill EM, Dyson L, Tildesley MJ, Keeling MJ (2021) Modelling optimal vaccination strategy for SARS-CoV-2 in the UK. PLOS Computational Biology 17(5): e1008849. <https://doi.org/10.1371/journal.pcbi.1008849>

2. *The vaccine drop-out rate modelled is 5%. I would like to see a sensitivity analysis on this value and especially in light of possible vaccine hesitancy towards ChAdOx1 nCoV-19 vaccine in different settings.*

Author response: The 5% vaccine dropout rate was obtained from the Voysey *et al.* paper² and conforms to the rates of refusal observed in the UK vaccination rollout³. Note that this is not a relevant parameter for our analyses (which is why we ignored it), since it is only a significant concern for younger people (not reached in our simulations given the limited dose allocation), and the core issue remains whether it would be better to vaccinate more people with a single dose or less people with two doses when the number of available doses is limited.

² Voysey, M., et al., Single dose administration, and the influence of the timing of the booster dose on immunogenicity and efficacy of ChAdOx1 nCoV-19 (AZD1222) vaccine Lancet, 2021.

³ <https://www.ons.gov.uk/releases/coronavirusandvaccinehesitancygreatbritain31marchto25april2021>

3. *Important characteristics of a vaccine against COVID-19 is the reduction in severity from vaccination – quantified by reduced hospitalisations and deaths- but also the impact of onwards transmission. The authors here assume that the vaccine efficacy against infection is 5% and they do not untangle the relationship between vaccine allocation and effectiveness of the vaccine on transmission. Decisions of rolling out vaccination in younger age groups, or in settings with younger population, is crucially depended on this and any informed evidence on vaccine allocation has to address this issue. My suggestion would be to include sensitivity analysis on this parameter and reproduce Figures 1-3 for different values of this parameter.*

Author response: As addressed in point 1, the ChAdOx1 nCoV-19 Phase 3 vaccine clinical trials found no statistically significant transmission-blocking effect (VE = 3.8% [-72.4 to 46.3]) both overall and at any stage of follow-up. We demonstrate in the methods section (Box 1) that for the low values of coverage explored here, the vaccine would need to have a very high infection blocking profile for indirect effects to be significant in our simulations, which would be against the evidence produced by the clinical trials. Also, there is no scope to vaccinate younger individuals (where indirect effects could play a larger role) since the vaccine prioritisation schedule was fixed to mimic the UK vaccine rollout (which was similar to most of the HICs), and thus, the potential for reductions in transmission was assumed to

be negligible throughout our simulations. Given that age is by far the most significant covariate to explain risk of deaths from infection, no country has of yet, decided to not prioritise the elderly population, since vaccinating them promptly is guaranteed to yield the most efficiency per dose. In addition, given that vaccine production rates are likely to be insufficient to meet the demand generated by a global pandemic in the short term, even if the vaccines have some infection prevention properties, the coverage of the vaccine in populations would not be able to reach the high rates needed to achieve significant transmission blocking.

Minor comments

- 5. Please give more detail on why the booster dose interval is following a specific beta distribution – and explain why variation of this is not considered in additional sensitivity analysis.***

Author response: We chose a beta distribution due to its appropriateness as a continuous probability distribution. With it we can easily define the proportion of individuals receiving the second vaccine dose a certain number of days after the first dose. We should note that this is not a random distribution but one that conforms with the second dose delay distribution in the ChAdOx1 nCoV-19 Phase 3 vaccine clinical trial. The distribution has a flat tail, i.e., delay times are spread out across the interval boundaries, which is realistic for a vaccine rollout programme where most people are likely to receive the second dose exactly X days after the first dose, but where others are likely to make other arrangements to receive the vaccine at their convenience over a longer time scale. We now provide a plot of the delay distribution for transparency – Figure S9.

Reviewer 2 COMMENTS:

- 6. The authors explain, "the methodology employed was very specifically tailored to the research question and its context." However, no source code is available for inspection from what I can tell. This is a serious issue for evaluating the model, for transparency and reproducibility. I always recommend against publication of papers based on models where the source code is not publicly available.***

Author response: The source code was made available in a GitHub repository as indicated in the paper under the heading “data sharing” as required by Nature Communications: <https://github.com/ricardoaguas/como-ChAdOx1-vaccine->. We can only assume this was not removed from the version the reviewer was provided.

- 7. The authors use an individual based model which in principle could be an unconstrained computer program that does anything, however there is a hint that it is much simpler than that: individuals are described by a tuple of attributes which are modified in a uniform way. This is good because it suggests that the model is just a stochastic rewriting system perhaps equivalent to a rule-based model or a coloured***

Petri net. I would find this more convincing than a tailored, bespoke, ad-hoc model because it means that we know how to understand the conceptual framework where it sits.

Author response: We thank the reviewer for this comment and agree that indeed the model framework used facilitates understanding of the processes involved and is invaluable for consistently comparing different scenarios assuming different attack rates. We should note that this is perhaps the most pertinent criticism the other reviewers have raised, so we really appreciate that this reviewer sees the value in the approach taken.

8. *The equation in "transmission and clinical cascade". Please number your equations. λ is a function age $\rightarrow \mathbb{R}$. However, age (a) does not appear on the right hand side. I think it perhaps corresponds to the "I" subscript in the numerator but as written this appears to be a constant vector-valued function of age which is clearly not what is intended.*

Author response: We thank the reviewer for the suggestion and have now numbered the equations in the methods section. However, we don't quite understand how there could be confusion regarding the force of infection equation. The text following the equation clearly defines c_{ij} as the daily number of contacts between age groups i and j for a particular country and P_j as the population age distribution. Since there are two age indices, we decided not to use the letter a (which is typically associated with age), in the right-hand side of the equation, to avoid any confusion. Clearly, we were not successful, and thus decided to change the notation based on this comment.

9. *However, this definition of λ is motivated by computational simplification because the simulation is expensive. It is unclear why the simulation should be so expensive. In particular, the authors say that the daily risk of infection is a function of time and is adjusted (how?) such that the correct attack rate is obtained. In the absence of a transmission model, what is the correct attack rate? How is this obtained? Surely it is itself influenced by vaccination which reduces the chance of infection. The mechanism here is unclear and is the main reason why I say that quantitative results obtained in this way should be taken with a grain of salt. Is it perhaps feasible, given that parameter regimes of interest have been identified using the relatively inexpensive model, to then explore those interesting regimes with a more fully-fledged model containing transmission? That would be my preferred approach over a static model.*

Author response: We considered a range of attack sizes (proportion of the population infected) during the six-month period which accounts for all non-pharmaceutical interventions in that timeframe. Mathematical models have been notoriously poor at predicting the future dynamics of the epidemic even in the medium-term. To avoid making extreme assumptions about what the future would look like, we opted to explore all possible realistic burdens within a 6 months'

time window, ignoring the minutia of the daily shape of the epidemic curve. To do that, we calibrate the model without vaccination to establish the daily risk of infection (same every day) for each level of attack rate – this is now clearly explained in the methods section. We then run model simulations with those daily risks of infection for the vaccine model and compare the outcome metrics of interest (deaths, clinical cases, and infections), thus establishing vaccine effectiveness as defined by equation (3) in the methods section. As mentioned in points 1 and 4 above, the vaccination campaigns simulated here can only have a very marginal impact on transmission since direct vaccine efficacy against infection is very low and there is a fixed rollout schedule that prioritised high-risk individuals who don't contribute very much to onwards transmission (Box 1). We have now also included Supplementary Figures 7 and 8 to clarify these effects.

10. Vaccine efficacy decays exponentially (next equation). This doesn't seem biologically realistic. At $t = 0$ I would expect efficacy to be zero and for it to increase to some maximum over a period of time and then to reach a plateau (probably not a sharp peak) and thereafter to slowly decay. This could matter because the time to reach the plateau is on a similar scale to the serial interval.

Author response: As indicated on page 9, we imposed a stepwise increase in post-dose two vaccine efficacy across an 8-week booster dose interval, as observed in the clinical trial¹. Furthermore, the trial revealed a slow decrease in protection following the first dose (in those that received one dose only) in agreement with an exponential decay. Note that the maximum interval between doses allowed in the model is 55 days, which is much lower than the lowest mean duration of vaccine protection explored.

¹ Voysey, M., et al., *Single dose administration, and the influence of the timing of the booster dose on immunogenicity and efficacy of ChAdOx1 nCoV-19 (AZD1222) vaccine* Lancet, 2021.

11. Ideally, I would like to see the effect of the vaccine on transmission incorporated here though I understand if the authors feel that it is out of scope. For countries with high rates of vaccination this is clearly important as we are already seeing in the USA and the UK. For countries that are severely supply constrained such as the LMICs that are really the main focus of this paper, this effect may not be so strong. At a minimum this should be discussed.

Author response: Please see our responses under 1, 4, and 9 above justifying the use of static model vs a transmission model. We have extended the methods section to provide more detail on the robustness of our assumptions and added some of those elements to the discussion as well, as suggested by the reviewer.

12. The observation of the threshold value for choosing between strategies corresponding to the curve whose integral is the proportion of the population over 65 mentioned in the caption of Fig 3 could usefully be emphasised. This seems to be the main result, actually, and it is mentioned almost in passing at the end of the Results section. However, it is not clear to me how the integral is defined because the curves are not functions (there are multiple "y" values for some "x"). Please clarify this and explain.

Author response: This integral is calculated from its *sensu strictu* definition of the area under the curve, the curve being the contour line with value 1. Note that these are surface plots, in which the contour line is defined by the mean of the Z values (relative vaccine effectiveness of single vs double dose regimens) for each combination of X and Y, as explained in Figure 2. This was outlined in the legend for Figure 3, but we have made the link clearer. Essentially if one dosing regimens were always superior to two dose regimens, the area under the curve would be 1, i.e., 100% of the surface plot area.

Reviewer 3 COMMENTS:

13. ChAdOx1 VE against circulating variants of concern, especially in LMIC settings where VOCs may cause the majority of cases.

Author response: The reviewer raises an interesting point that is, however, outside the scope of this paper. We address the optimal logistical deployment of a vaccine campaign based on the immunological profile of the ChAdOx1 vaccine. Given that data on ChAdOx1 vaccine efficacy results against VOCs are speculative at best, these have not been included. We did however consider that efficacy could potentially be lower than the estimate presented in the original clinical trial, thus accounting for some viral immune escape over time. Note that the lowest vaccine efficacy values explored in the sensitivity analysis are 32.5% after the first dose and 65% after the second dose, which constitutes a 24-50% decrease in efficacy from the best-case efficacy scenario.

14. National regulatory agencies' decision-making process regarding ChAdOx1 vaccine safety in younger populations, and how this could influence age-based prioritization and/or speed of rollout. Alternate scenarios in which (usually younger) healthcare workers are prioritized for vaccination alongside older adults; this may be of special interest given SAGE recommendations for vaccine prioritization and given new safety outcomes of interest in Europe.

Author response: We do not address that issue as there was no scope to vaccinate younger individuals (where indirect effects could play a larger role) since the vaccine prioritisation schedule was fixed to mimic the UK vaccine rollout (which is similar to most of the HICs). Crucially, this work tries to provide insights into the potential value of vaccinating those

at higher risk of death with a single dose vs a double dose under very strict dosing availability scenarios. Additionally, we consistently argue that direct vaccine protection against deaths is dominant unless a very large proportion of the younger population (responsible for most infectious contacts) could be targeted. Other modelling groups have tackled the age prioritisation issue and have reached the same conclusion¹ – that the highest risk group (elderly) should always be prioritised first if the aim is to reduce the mortality burden.

¹ Moore S, Hill EM, Dyson L, Tildesley MJ, Keeling MJ (2021) Modelling optimal vaccination strategy for SARS-CoV-2 in the UK. PLOS Computational Biology 17(5): e1008849. <https://doi.org/10.1371/journal.pcbi.1008849>

Additional minor comments are listed below.

15. Page 4, paragraph 1: It would be helpful for the purposes of contrast to also list the estimated efficacy for ChAdOx1, since it is listed for the Pfizer and Moderna vaccines, and to specify the outcomes for which efficacy was measured (for all vaccines)

Author response: We did not include this as this vaccine estimated efficacy is one of the parameters explored in our sensitivity analysis. Granted, in our sensitivity analysis we generate uncertainty around the estimate provided in the OVG paper (Voysey *et al*)², so we have taken the reviewers suggestion and added this to the introduction/background section of the paper.

² Voysey, M., et al., *Single dose administration, and the influence of the timing of the booster dose on immunogenicity and efficacy of ChAdOx1 nCoV-19 (AZD1222) vaccine* Lancet, 2021.

16. Page 4, paragraph 2: As a point of clarification, when the authors say “suffer complex vaccine responses”, are they referring to vaccine safety/adverse event outcomes?

Author response: We thank the reviewer for picking up on this poor choice of wording. The statement refers to immunogenicity of the vaccine in populations with high levels of malnutrition and other infectious diseases’ incidence. This has been clarified in the text.

17. The authors note a hospitalization fatality rate; perhaps they could note which proportion of COVID-19 deaths in low-resource settings they assume to occur outside the hospital.

Author response: This is an interesting point that we have tackled continuously with in other work (see

<https://como.bmj.com/>). Ultimately, we have found that deaths tend to be under-reported in LMICs, but that the magnitude of that bias is very difficult to quantify. Here, we assume that differences in fatality across countries are only a reflection of their age structure, rather than differences in population frailty. Also note that the numbers of infections, cases and deaths in our results pertain to “true” outcomes and not what would be reported by the health system. In conclusion, we do not take country level access issues into account beyond age structure and “availability” as defined by the maximum population that would be covered by COVAX.

18. Page 9, paragraph 5: By “an individual with baseline status i”, do the authors mean “baseline vaccination status”?

Author response: We adopted the standard nomenclature used in vaccine trials, defining individuals as previously exposed or naïve at enrolment (baseline). The baseline status is then the susceptibility status of population (immune vs. non-immune) before the vaccination campaign starts. This has been made clearer in the methods section.

19. At the end of the discussion section, it could be helpful to briefly mention which additional observational data (e.g., VE estimations from Canada which will now include a 4-month delay between dose 1 and dose 2) will be helpful to update these calculations

Author response: We agree that including these elements is a valuable addition and have made the appropriate changes in the discussion.

20. In Figure 1, is the top bar x-axis representing VE? If so, it would be helpful to provide that information in the axis title or labels. Similarly, it can be challenging to flip back and forth between Table 1 and Figure 1; it would be helpful to include the abbreviations in row titles in Figure 1 as a footnote as well.

Author response: We are unsure what the reviewer means by “top bar x-axis”. The figure legend and title clearly mention that the boxplots show the median and interquartile ranges of the predicted vaccine effectiveness (VE) on each of the outcomes (Deaths, Clinical cases, and Infections). We fully agree that adding the abbreviations as a footnote might facilitate reading and have made the appropriate changes.

21. In Figure 2, if I understand the figure correctly, I recommend the vertical bar on the right side be removed and rotated 90 degrees so that it reads more as a “legend” than as a secondary y-axis. (Again, if I do not understand correctly, my apologies.)

Author response: The vertical bar is the colour legend. This is a standard presentation of a surface plot, with a figure legend on the right-hand side. We have left an empty space between the plot and the colour-bar to avoid any confusion.

22. In Figure 2, it is not clear to me what the decimal represents for “dose allocation”—does this represent the percentage of the population vaccinated, the percentage of total doses that are given in a 6 month time frame, or other?

Author response: This is simply a different numerical representation of a percentage value (0.3 = 30%). As defined in the *Vaccination delivery and vaccine efficacy* section of the Methods section and in Table 1, this represents the allocation range, i.e., the number of doses available as a percentage of the total population. For example, for a population of 10 million people that has 1 million doses available, that would correspond to 10% dose allocation.

REVIEWER COMMENTS

Reviewer #1 (Remarks to the Author):

Overall, the authors have taken in consideration my comments and responded adequately to the points I have raised. I am happy to recommend acceptance of this paper for publications following this revision.

Below are couple of suggestions to improve the timeliness of this work in light of the ongoing vaccine rollout:

1) An extra analysis that would be nice to include is additional third dose of the vaccine - as a booster vaccine. Exploring whether this additional vaccine would improve outcomes would be interesting, If the authors feel this is better suited to future analysis, maybe they can add a paragraph on this in the discussion instead.

2) Updating the model parameters for effectiveness of this vaccine on onwards transmission - in light of recent PHE work that onwards transmission can be reduced by 49% in vaccinated people with this vaccine

<https://khub.net/documents/135939561/390853656/Impact+of+vaccination+on+household+transmission+of+SARS-COV-2+in+England.pdf/35bf4bb1-6ade-d3eb-a39e-9c9b25a8122a?t=1619601878136>- would also be a useful discussion point.

Reviewer #2 (Remarks to the Author):

I thank the authors for their detailed responses. Whilst I still believe that a dynamic transmission model would be better, I accept that it was the authors' express intention to implement a static one due to a lack of evidence for transmission blocking effects of vaccine and the focus being on low resource settings where coverage can be expected to be low. All of my concerns have been addressed except one, as well as a minor point.

EQ. 1 – FORCE OF INFECTION, $\lambda(a)$

Also thanks for adjusting the notation for $\lambda(a)$ (Equation 1). It is now clearer though I still have a minor niggle. The authors write,

$$\lambda(a) = k_\lambda \sum_{w=1}^N c_{aw} \frac{\sum_{a=1}^N c_{aw}}{\sum_{a=1}^N \sum_{w=1}^N c_{aw}}$$

this is ambiguous because the a that represents the argument to the function is only the a in the first c_{aw} term and is not the same a as appears in the sums. To fix this, just choose a different symbol in the sums, e.g.,

$$\lambda(a) = k_\lambda \sum_{w=1}^N c_{aw} \frac{\sum_{v=1}^N c_{vw}}{\sum_{v=1}^N \sum_{w=1}^N c_{vw}}$$

FIG. 3 – AREAS UNDER CURVES

The authors respond that they are computing the area under the curves in this figure. I queried what this could mean because the curves are not functions so the integral is not defined, and the authors replied that they are using the definition of an integral that means the area under the curve. This is still not satisfactory. I have annotated the figure to help explain the problem. The usual way of

understanding area under the curve is to take the limit of a sum of small area elements, say of width

2

δ .

$$\int_{0.5}^1 f(x)dx = \lim_{\delta \rightarrow 0} \sum_{n=0}^{\frac{0.5}{\delta}} [f(0.5 + (n+1)\delta) - f(0.5 + n\delta)] \delta$$

The problem is, what is the value of $f(x)$ and $g(x)$ where they intersect the red line? $f(x)$ would have *three* distinct values and $g(x)$ would have five. That means that the sum above is not defined and neither is the area.

Based on the authors' response, I *think* what is meant is the fraction of the surface area enclosed by the curves (and, implicitly, both axes). This makes sense but is a very different concept from area under the curve. For this to make sense, the treatment of the two excursions (circled in red) needs to be specified. I believe that the authors mean the area enclosed by the main $h(x)$ curve and the axes, minus the area of the small excursions. This could be written as,

$$\int_0^{0.3} \int_{0.5}^1 f'(x, y) dx dy$$

where,

$$f'(x, y) = \begin{cases} 1 & f(x, y) \leq 1 \\ 0 & \text{otherwise} \end{cases}$$

and I have taken y to be the "Dose allocation" axis.

If this is right, the general idea would be better conveyed as "the area enclosed by the curves". Writing what is meant in math, correctly, is good to make sure the meaning is clear. The term "area under the curve" should not be used because (a) it is incorrect as I show above and (b) it is very confusing as this exchange shows.

Reviewer #3 (Remarks to the Author):

I thank the authors for the time spent revising this article, which I do believe has important value for vaccine implementation programs. However, there are challenges to the manuscript interpretation which still remain after revision, and which would be good to address before the manuscript is published. These fall into two main categories, described below:

1. Considerations for other environments/epidemiological contexts. The authors have made an effective argument for the analysis they have completed, and acknowledge that adjustment of existing parameters or addition of new ones—while likely helpful for different contexts—is outside the scope of the paper. I understand this, but would then appreciate more mention in the discussion of how these factors could change policy decisions based on their findings and other literature that has been published in the meantime. The authors noted that calculated VE for ChAdOx1 against variants of concern is “speculative at best”, which may have been true when the response was written; but they will hopefully be able to provide preliminary estimates from Public Health England and other groups at this point.

2. Interpretation of figures and tables. The authors have noted that the figures are standard and should be able to be interpreted easily, but unfortunately, the paper will be most effective if the figures are explained to the lowest common denominator (a group which certainly must include myself). It is still unclear to me how to interpret the third column in Figure 1, but it appears to show an estimated VE against SARS-CoV-2 infections of 0.4%. This seems low, given existing literature on this topic. Similarly, it would be helpful to change the y-axis title in Figure 2 to the more accurate phrase “Dose allocation – maximum doses available per population”. The current description, “number of doses available,” is not strictly accurate; the value shown is not a number but a proportion.

Author's note:

We appreciate the reviewers' constructive comments and feel the revised manuscript should clarify all the concerns raised and be a lot clearer to a wider audience. All new sources of data and code used to generate the results in the detailed response to reviewer's comments below can be found in the github repository that accompanied the original submission: <https://github.com/ricardoaguas/como-ChAdOx1-vaccine->.

Reviewer 1

1.1. An extra analysis that would be nice to include is additional third dose of the vaccine - as a booster vaccine. Exploring whether this additional vaccine would improve outcomes would be interesting, if the authors feel this is better suited to future analysis, maybe they can add a paragraph on this in the discussion instead.

Author response: We appreciate the reviewer's positive feedback and are very happy to respond to their comments. Recent discussions have focused on the potential population-level impact of an additional third booster dose of the vaccine, with Israel even implementing a third round of vaccinations without FDA approval. We feel that the lack of clinical data on the potential additional benefits of a third dose warrants further research especially given the uncertainty in the duration of the vaccine's protective effectiveness following the second dose. Additional data from trials would enable us to conduct post hoc analyses on optimal dosage strategies including a third dose in a variety of demographic settings. We have included this additional commentary in the discussion section of the manuscript.

1.2. Updating the model parameters for effectiveness of this vaccine on onwards transmission - in light of recent PHE work that onwards transmission can be reduced by 49% in vaccinated people with this vaccine <https://khub.net/documents/135939561/390853656/Impact+of+vaccination+on+household+transmission+of+SARS-COV-2+in+England.pdf/35bf4bb1-6ade-d3eb-a39e-9c9b25a8122a?t=1619601878136> would also be a useful discussion point.

Author response: We thank the reviewer for this observation. We have been attentively following the literature on the population level impact of vaccination rollout, with a special interest in Israel and the UK. In the UK there have been a series of epidemiological studies trying to determine how community and household transmission is being modulated by vaccination^{1,2}. These studies suffer from a series of sampling and frame of reference issues which result in the highest level of protection from the vaccine being found in the days immediately following vaccination – when antibody levels induced by vaccination are known to be negligible³.

¹ <https://www.nejm.org/doi/full/10.1056/NEJMc2107717>

² <https://www.medrxiv.org/content/10.1101/2021.03.11.21253275v1>

³ [https://www.thelancet.com/journals/lancet/article/PIIS0140-6736\(20\)31604-4/fulltext#seccestitle150](https://www.thelancet.com/journals/lancet/article/PIIS0140-6736(20)31604-4/fulltext#seccestitle150)

Regardless, we do believe an argument can be made for lower infectivity of vaccinated individuals as a consequence of lower viral loads following infection (we still assume risk of infection following vaccination is largely unchanged). Thus, we now calculate a daily modulator (\widehat{vb}) of the risk of infection that accounts for the vaccine impact on onwards transmission depending on the proportion of people in the population with j doses of the vaccine:

$$\widehat{vb}(t) = 1 - \frac{\sum_{i=0}^{i=N} v_{impact}^j}{N}$$

Consistent with the remaining vaccine efficacy parameters, we assumed there is a boost in vaccine impact on transmission with increasing number of doses given by parameter D2B. In plain terms, under homogeneous transmission assumptions, the overall impact on transmission of vaccinating a proportion of the population is equal to the mean decrease in transmission across all individuals. To circumvent this limitation, we assume that the mean impact on transmission changes daily to reflect how the network of contacts on a given day might contain different proportions of vaccinated people. This is done by sampling a population level impact on transmission vb assuming a Beta distribution with overdispersion σ :

$$vb(t) \sim \beta(\widehat{vb}(t), \sigma)$$

This parameter vb changes daily as more people get vaccinated and modulates the population force of infection ($\lambda(a)$) directly:

$$\lambda(a, t) = \lambda(a) \cdot vb(t)$$

We performed similar sensitivity analyses as the ones in the main manuscript that include vaccine impact on infectivity and retrieve the results in **Figure 1** below.

The methodological description above along with the additional sensitivity analysis (with maximal values for vaccine impact on infectivity of 25% and 50%) are now provided in the supplementary materials (Supplementary Figure 10) and referenced to in the discussion.

Figure 1. Sensitivity analysis mimicking that in Figure 1 of the previous submission, now including a max vaccine effect on transmission given by parameter RT.

2. Reviewer 2

2.1. Eq. 1 (Force of infection, $\lambda(a)$). The a that represents the argument to the function is only the a in the first c_{aw} term and is not the same a as appears in the sums. To fix this, just choose a different symbol in the sums.

Author response: We thank the review for calling our attention to this potential source of confusion and have made the appropriate changes as shown below and on page 8.

$$\lambda(a) = k_{\lambda} \sum_{w=1}^N c_{aw} \frac{\sum_{v=1}^N c_{vw}}{\sum_{v=1}^N \sum_{w=1}^N c_{vw}}$$

2.2. Fig. 3 – Areas under curves. The authors respond that they are computing the area under the curves in this figure. I queried what this could mean because the curves

are not functions so the integral is not defined, and the authors replied that they are using the definition of an integral that means the area under the curve. This is still not satisfactory.

Author response: We apologize for not being clearer in our response. To be very specific, we take the contour lines of the linearly interpolated outcome values for each combination of x and y values. We then vectorize each contour line to obtain the y-values at which the contour line is crossed for evenly spaced x-values (taking the highest y-value for duplicates). The resulting vector is then run through a composite trapezoid rule algorithm that computes the area under the curve (auc function in the MESS R package). This is now made more explicit in the caption for figure 3 in the manuscript.

3. Reviewer 3

3.1. Considerations for other environments/epidemiological contexts. The authors have made an effective argument for the analysis they have completed and acknowledge that adjustment of existing parameters or addition of new ones—while likely helpful for different contexts—is outside the scope of the paper. I understand this but would then appreciate more mention in the discussion of how these factors could change policy decisions based on their findings and other literature that has been published in the meantime. The authors noted that calculated VE for ChAdOx1 against variants of concern is “speculative at best”, which may have been true when the response was written; but they will hopefully be able to provide preliminary estimates from Public Health England and other groups at this point.

Author response: We appreciate the reviewer’s positive feedback. We are cognisant of the influence of diverse environments and epidemiological settings on vaccine effectiveness. However, to fully evaluate the specific impacts, additional research is needed on the implications of the epidemiological and health system contexts - particularly in LMICs. We have updated the discussion, highlighting the importance of understanding how the impact of vaccinating specific population sub-groups is a product of a trade-off between the relative proportion of those at highest risk vs. those that contribute the most to onwards transmission which is balanced by the magnitude of the impact of vaccination on transmission (page 12, second paragraph).

We maintain that our intention in this study is to investigate optimal dosing strategies for a specific vaccine delivery strategy (vaccinating the high-risk groups first) under different dose availability and population structure constraints. However, we have expanded the discussion to provide examples of ongoing studies addressing other health contexts and related themes, and how to interpret our results in light of those studies.

3.2. Interpretation of figures and tables. The authors have noted that the figures are standard and should be able to be interpreted easily, but unfortunately, the paper will be most effective if the figures are explained to the lowest common denominator (a group which certainly must include myself). It is still unclear to me how to interpret the third column in Figure 1, but it appears to show an estimated VE against SARS-CoV-2 infections of 0.4%. This seems low, given existing literature on this topic. Similarly, it would be helpful to change the y-axis title in Figure 2 to the more accurate phrase “Dose allocation – maximum doses available per population”. The current description, “number of doses available,” is not strictly accurate; the value shown is not a number but a proportion.

Author response: We should stress that the analyses contained in the original submission assumed that the vaccine infection blocking effect was at most 5% at the individual level. As a consequence, the population level impact on infections of vaccinating at most 30% of people with a single dose is negligible. However, in the revised manuscript we now provide an additional analysis accounting for vaccine induced reduction in infectivity (Supplementary Figure 10). Please refer to our response to point 1.2 above for a comprehensive description of the methodology.

Regarding the second half of the comment, we appreciate the suggestion and have changed the axis label in Figure 2 to “Dose allocation – maximum doses available per population”.

REVIEWERS' COMMENTS

Reviewer #1 (Remarks to the Author):

I am very pleased that the authors have successfully addressed both of my comments and I am content to recommend acceptance of this manuscript.

On the first comment: adding a discussion paragraph on the addition booster vaccine, while awaiting further results from trials, suffices my first comments.

On the second comment: The extra analysis that they have undertaken on the boost of vaccine efficacy with an increasing number of people in the population being vaccinated and hence reliant on the population network structure, together with the sensitivity analysis of 25% and 50% vaccine effect on infectivity, are very good additions to the paper methods. These two parts have sufficiently answered my query.

One additional point that the authors could add to the discussion is the avenue of further work where this analysis is complemented with analysis looking at waning of immunity from vaccination: both the time and the shape of the distribution and how this differs in a one-dose, two-dose and three-dose regime.

Reviewer #2 (Remarks to the Author):

I thank the authors for taking the time to address my concerns. All of my concerns have been addressed and I am happy to recommend the article for publication.

Author's note:

We appreciate the reviewers' constructive comments and feel the revised manuscript should clarify all the concerns raised and be a lot clearer to a wider audience. All new sources of data and code used to generate the results in the detailed response to reviewer's comments below can be found in the github repository that accompanied the original submission: <https://github.com/ricardoaguas/como-ChAdOx1-vaccine-> (doi:10.5281/zenodo.5522794).

Reviewer 1

Comment: I am very pleased that the authors have successfully addressed both of my comments and I am content to recommend acceptance of this manuscript.

1.1. On the first comment: adding a discussion paragraph on the addition booster vaccine, while awaiting further results from trials, suffices my first comments.

We thank the reviewer for acknowledging the addition.

1.2. On the second comment: The extra analysis that they have undertaken on the boost of vaccine efficacy with an increasing number of people in the population being vaccinated and hence reliant on the population network structure, together with the sensitivity analysis of 25% and 50% vaccine effect on infectivity, are very good additions to the paper methods. These two parts have sufficiently answered my query.

We thank the reviewer for acknowledging the addition.

1.3. One additional point that the authors could add to the discussion is the avenue of further work where this analysis is complemented with analysis looking at waning of immunity from vaccination: both the time and the shape of the distribution and how this differs in a one-dose, two-dose and three-dose regime.

We have included this addition in the last paragraph of the discussion.

Reviewer 2

Comment: I thank the authors for taking the time to address my concerns. All of my concerns have been addressed and I am happy to recommend the article for publication.

We thank the reviewer for acknowledging the revisions and for recommending the article for publication.